# The effect of methylphenidate on anaesthesia recovery: An experimental study in pigs

**Alessandro Mirra**[ID][1]*, **Fabiana Micieli**[ID][2], **Mirjam Arnold**[3], **Claudia Spadavecchia**[1], **Olivier Louis Levionnois**[ID][1]

1 Section of Anaesthesiology and Pain Therapy, Department of Clinical Veterinary Medicine, Vetsuisse Faculty, University of Bern, Bern, Switzerland, 2 Department of Veterinary Medicine and Animal Production, University of Naples "Federico II", Naples, Italy, 3 Clinic for Swine, Department for Clinical Veterinary Medicine, Vetsuisse Faculty, University of Bern, Bern, Switzerland

* alessandro.mirra@vetsuisse.unibe.ch

## Abstract

### Introduction

Due to the lack of specific antagonists for general anaesthetics, the pharmacological stimulation of the arousal pathways might contribute to reduce recovery time. We aimed at assessing the effect of methylphenidate on physiological parameters, nociceptive withdrawal reflex thresholds, electroencephalographic variables and time of reappearance of reflexes in pigs undergoing propofol anaesthesia.

### Materials and methods

Two experiments have been performed. Five (experiment 1) and sixteen (experiment 2) healthy juvenile pigs were anaesthetised with propofol. In experiment 1, saline, methylphenidate 10 mg/kg or methylphenidate 20 mg/kg was administered intravenously at the end of propofol administration, using a cross-over design. In experiment 2, saline (n = 8) or methylphenidate 20 mg/kg (n = 8) was administered immediately after extubation. In both experiments, physiological parameters, nociceptive withdrawal reflex thresholds, electroencephalographic variables and time of reappearance of reflexes were assessed. Comparison among groups was performed using either the two-way repeated measures ANOVA followed by Bonferroni-Test or the t-test in case of parametric data, and either the Kruskal-Wallis test or the Mann-Whitney Rank Sum test in case of non-parametric data. A p value < 0.05 was considered statistically significant.

### Results

No clinically relevant changes were observed in both experiments for physiological parameters, nociceptive withdrawal reflex thresholds and electroencephalographic variables.

### Conclusions

Methylphenidate does not shorten or modify anaesthesia recovery in pigs, when the sole propofol is administered.

**Data Availability Statement:** Relevant data are within the manuscript, its Supporting Information files, and in the Zenodo database (https://doi.org/10.5281/zenodo.10018392).

**Funding:** This work was supported by the Swiss National Science Foundation (Spark) [https://www.snf.ch/en. Grant number CRSK-3_190926/1. Main grant applicant: OL], the Berne University Research Foundation and BEKB Förderfonds [https://forschungsstiftung.ch/. Grant number 22/2020. Main grant applicant: OL] and by the Faculty Clinical Research Platform (FKFP) of the Vetsuisse Faculty of the University of Bern [Main grant applicant: AM]. The funder had no role in study design, data collection and analysis, decision to publish, or preparation of the manuscript. There was no additional external funding received for this study.

**Competing interests:** The authors have declared that no competing interests exist.

## Introduction

While induction of general anaesthesia is actively performed via drugs administration, recovery from the anaesthetised status relies mostly on awaiting their metabolization and clearance. The lack of control during this delicate phase leads to increased risks for the patient, not only related to unconsciousness, but also to respiratory depression and lack of protective reflexes (e.g., laryngeal reflex) [1, 2]. The economic burden associated with prolonged recovery highlights the need to shorten it without quality loss [3, 4].

Since mechanisms of action and receptors targeted by general anaesthetics are not entirely elucidated, it is challenging to develop drugs capable of reversing their effects. A possible alternative is the less specific stimulation of arousal pathways. Different neurotransmitters are involved in their activation (e.g., noradrenaline, dopamine, acetylcholine), but none has been shown to play a pivotal role. So far, the most promising results have been obtained using methylphenidate (MP), an inhibitor of dopamine and norepinephrine reuptake transporters [1, 5–7]. Its capacity of counteracting the effects of barbiturates in different species has been demonstrated long ago [8–10]. Subsequent studies in rats have shown its ability to decrease recovery time after a single propofol bolus, and to induce emergence during continuous propofol or isoflurane administration [5, 6]. Moreover, previous reports demonstrated that MP can counteract respiratory depression, reduces intubation time and facilitates return to consciousness [11–13].

Despite there has been a renewed interest for this drug in recent years, definite conclusions about its efficacy have yet to be drawn. Moreover, dose and timing at which MP should be administered to shorten recovery time and minimize adverse effects has not been established.

The present study consisted of two consecutive experiments: experiment 1, aiming at assessing the effect of two different doses of MP (10 or 20 mg/kg IV) on anaesthesia recovery time and characteristics, when administered at discontinuation of propofol infusion; experiment 2, aiming at assessing the effect of MP at the dose showing the best results in experiment 1, on a larger population.

We hypothesised that MP, administered at the selected doses, would shorten recovery time without inducing side effects.

## Materials and methods

Ethical approval (protocol number 32015) was obtained from the Committee for Animal experiments of the Canton of Bern, Switzerland. The experiments were conducted between June 2020 and May 2021.

### Animals

For experiment 1, five healthy pigs (phenotype Edelschwein), 10.4 ± 0.5-week-old (mean ± standard deviation), weighting 25.8 ± 1.8 kg, and of mixed sex (one female and four castrated males) were included. Each pig was anaesthetized three times following a cross-over design, ensuring a wash-out period of at least 36 hours between sessions. At the end of the third session, pigs were euthanised with IV pentobarbital (150 mg/kg).

For experiment 2, sixteen healthy pigs (phenotype Edelschwein), 10.0 ± 0.6-week-old, weighting 28.9 ± 4.7 kg, and of mixed sex (ten females and six castrated males) were included. At the end of the experiment, pigs were euthanised with IV pentobarbital (150 mg/kg).

All the animals were collected from the farm of origin between two and ten days before the beginning of the experiment and transported to the animal facility of the University of Bern, Vetsuisse faculty. At least two pigs at a time were kept in the facility, with visual and auditory contact continuously guaranteed.

## Preparation and clinical monitoring

On the experimental day, one pig was brought into the experimental room and allowed to acclimatize for approximately 30 minutes before being placed into an on-purpose made sling. A local anaesthetic cream (EMLA, 5%, Anesderm, Pierre Fabre, Switzerland) was applied to the auricular vein and artery, and over the coccygeal artery, ensuring complete coverage, and left acting for at least 45 minutes before catheters placement. During this period, a six leads wireless device (Televet 100, Engel Engineering Services GmbH, Germany) was applied to continuously record electrocardiographic activity. The area between the frontal and the occipital bone, over the tibialis cranialis muscle and over the right metatarsus was clipped, cleaned with a warm antiseptic soap solution and shaved with a razor. Once dried, the skin was rubbed with abrasive paper (Red Dot Trace Prep, 3M Health Care, Canada) and benzinum medicinale (Benzinum Medicinale, Hänseler AG, Switzerland) was applied to the skin to remove excess fat. During the whole experimental period, the following clinical variables were continuously assessed and reported on an anaesthesia record sheet every 5 minutes: heart rate (HR), respiratory rate (RR), diastolic (DAP), mean (MAP) and systolic (SAP) invasive blood pressure, as well as palpebral reflex and jaw tone, that were scored as "0" = absent, "1" = reduced or "2" = as awake.

## Nociceptive withdrawal reflex (NWR) equipment and threshold determination

A dedicated unit (PainTracker; Dolosys GmbH, Germany) was used to continuously assess the nociceptive withdrawal reflex thresholds (NWRt) at the right hind limb. Two surface electrodes were placed over the common dorsal digital nerve for transcutaneous electrical stimulation, and two surface electrodes were placed over the tibialis cranialis muscle on the same limb to record electromyographic activity (EMG). A ground electrode was placed in proximity. Electrical stimulation was repeated every 10 seconds with 50% randomization. Each stimulation consisted of five rectangular pulses of 1 ms duration delivered at 200 Hz. The stimulation intensity was started at 1 mA and then automatically adapted using a bracketing design, according to the EMG response obtained (positive or negative). Step size was set at 0.3 mA but increased to 0.5 mA when the intensity had changed in the same direction for the three last consecutive measurements. The time window considered for analysis of the EMG response (reflex) was set between 40 and 140 ms following stimulus onset (NWR range), while the window between 130 and 10 ms prior to stimulation onset was assessed to quantify the presence of noise (noise range). Measurements were discarded when the EMG amplitude exceeded 15 microvolts ($\mu V$) within the noise range; in this case, the stimulus was repeated at the same intensity. The reflex was considered positive at an interval peak Z-score above 10, meaning that the difference between the maximum EMG amplitude within the NWR range and the mean EMG amplitude within the noise range had to be above 10-fold the standard deviation of the EMG amplitudes within the noise range. The NWRt was automatically estimated after each stimulation through a logistic regression of the last 12 valid stimuli. During the whole experimental period, the NWRt was continuously tracked and reported on an anaesthesia record sheet every 5 minutes.

## Electroencephalographic signal recording

A pediatric RD SedLine® EEG-sensor was positioned as previously described [14]. Briefly, the electrodes were placed on a transverse line over the frontal bone, keeping their rostral border on an imaginary line running between the lateral canthi of the eyes. The central CB

(ground) and the caudal CT (reference) electrodes were placed on the mid-sagittal line. The SedLine®-display was set at 10 µV/mm and 30 mm/sec. The following SedLine®-generated variables were continuously assessed: Patient State Index (PSI), spectral edge frequency 95% right (SEF r) and left (SEF l), suppression ratio (SR, in %), electromyographic activity (EMG, in %) and presence of artifact (ART, in %). In addition to the SedLine monitor, a second EEG device consisting of an amplifier (EEG100c, Biopac Systems Inc, CA, USA), a data acquisition module (MP160, Biopac Systems Inc, CA, USA) and related software (AcqKnowledge, Biopac Systems Inc, CA, USA) was used to continuously record the raw electroencephalographic activity. For this purpose, four EEG surface electrodes (Ambu® Neuroline 715; Ambu, Ballerup, Denmark) were placed caudally to the SedLIne electrodes. Left (L-) and right (R-) electrodes were placed symmetrically, lateral to the mid-sagittal line, on the same sagittal line than L1 and R1, just rostral to the caudal margin of the occipital bone (L-pos, R-pos), and in the middle between these and the RD SedLine® EEG-sensor (L-mid, R-mid). The data from the Biopac (R-mid) were used to analyse the electroencephalographic power. During the whole experimental period, SedLine®-generated variables were continuously recorded and reported on an anaesthesia record sheet every 5 minutes.

## Treatments and statistical analysis

**Experiment 1.** Experiment 1 was developed as an experimental, randomized, blind, cross-over study. Each pig underwent three treatments, one for each session. Treatments were assigned in a random order (www.randomizer.org). After at least five minutes of pre-oxygenation via a face mask, propofol (Propofol 1% MCT, Fresenius Kabi AG, Switzerland) was administered to induce general anaesthesia as follows. A continuous propofol IV infusion was started at 20 mg/kg/hour and a first IV bolus was administered (4–5 mg/kg) followed by smaller doses (0.5–1 mg/kg), repeated every 30–60 seconds. Attempts to intubate the trachea were performed until absence of reflex responses allowed successful intubation. Then, 15 minutes were waited to allow stabilization of the propofol plasmatic concentration. The infusion rate was then increased by 6 mg/kg/hour every ten minutes, together with an additional IV bolus of 0.5 mg/kg. The pigs were always let breathing spontaneously. The propofol infusion was discontinued once SR (as calculated by the SedLine® monitor) reached a value between 10% and 30% and was stable for 10 consecutive minutes. Then, one of the three randomly assigned treatment was administered IV: saline (SL), methylphenidate 10 mg/kg (MP10) or methylphenidate 20 mg/kg (MP20). The investigator injecting the treatment was blind to it.

Clinical variables, NWRt, SedLine®-generated variables and raw EEG data collected from the R-mid electrode from the Biopac system, and times to recovery were analysed for a potential effect of MP. The SedLine®-generated data were downloaded from the monitor in an Excel sheet. Electroencephalographic power was assessed for the following frequency bands: total (0.1–35 Hz), delta (0.1–4 Hz), theta (5–8 Hz), alpha (9–12 Hz), beta (13–25 Hz), and gamma (26–35 Hz). Calculated values of total, delta, theta, alpha, beta and gamma EEG power, as well as PSI, SEF l, SEF r, SR, EMG, ART and NWRt were averaged over one minute. Parameters were assessed at the following time points:

- Baseline: 5 minutes before treatment;

- TP5: 5 minutes after treatment;

- TP10: 10 minutes after treatment;

- TP15: 15 minutes after treatment.

Different time spans were also calculated and further analysed; namely, time to: a. Extubation; b. Return to normal palpebral reflex; c. Return to normal jaw tone; d. Return to NWRt baseline; e. PSI 50; f. PSI 60; g. PSI 70; h. PSI 80.

Normality was assessed using a Shapiro-Wilk test. Comparison among groups were performed using the two-way repeated measures ANOVA followed by Bonferroni-Test in case of parametric data and the Kruskal-Wallis test in case of non-parametric data. A p value < 0.05 was considered statistically significant. Due to the lack of previous studies on the effect of MP on anaesthesia recovery in pigs, sample size was calculated using the Resource Equation Approach [15], considering 3 groups.

### Experiment 2

Experiment 2 was designed as an experimental, randomized, blind study. After at least five minutes of preoxygenation via a face mask, propofol infusion was started at a rate of 10 mg/kg/hour and increased by 10 mg/kg/hour every 15 minutes. Oxygen supplementation was continuously provided via face mask. When deemed appropriate (or necessary) by the anaesthetist (AM), gentle attempts at endotracheal intubation were made. After successful intubation, volume-controlled mechanical ventilation was started (tidal volume = 15 ml/kg; respiratory rate adjusted to reach an end-tidal carbon dioxide ($EtCO_2$) between 35 and 45 mmHg). Once SR remained above 80% for 10 consecutive minutes, propofol infusion was stopped and the pig allowed to recover. Tracheal extubation was performed as soon as signs of recovery, swallowing and chewing movements were noticed (as done in routine clinical practice). Immediately after extubation, one of the two randomly assigned treatment (www.randomizer.org) was administered IV: saline (SL_2) or Methylphenidate 20 mg/kg (MP_2). The investigator injecting the treatment was blind to it.

The same variables and time points described for experiment 1 were evaluated. Normality was assessed using the Shapiro-Wilk test. Comparison among groups were performed using the t-test in case of parametric data and the Mann-Whitney Rank Sum test in case of non-parametric data. A p value < 0.05 was considered statistically significant.

Sample size has been calculated considering the difference in recovery time between groups when administering either MP or saline. It was based on experiment 1, in which pigs needed 72.8 ± 14.7 and 43.6 ± 18.5 minutes for returning to a PSI of 80 while receiving saline or MP at 20 mg/kg, respectively. Considering an alpha of 0.05 and a power of 0.80, 8 animals per group were required.

## Results

For both Experiments, when data downloaded from the SedLine® were not available (error in recording) the values reported on the anaesthesia sheet were used (more details can be found in the supplemental material and in the files in the repository: https://doi.org/10.5281/zenodo.10018392).

### Experiment 1

Significantly higher baseline values were found for RR in Group MP10 compared to Group SL (p = 0.043), and for DAP (p = 0.022) and MAP (p = 0.046) in Group MP20 compared to Group SL. Significantly higher MAP was observed in Group MP10 compared to Group SL at TP10 (p = 0.042), and higher SAP at TP15 (p = 0.049). Moreover, a significantly longer time was needed to regain the jaw tone in Group MP20 compared to Group SL (p = 0.015), and to reach a PSI of 80 in Group SL compared to Group MP10 (p = 0.039). No other significant differences among groups were found. Details are reported in the S1 Appendix.

### Experiment 2

Before treatment administration, palpebral reflex and jaw tone were already present in 11/16 and 13/16 animals, respectively; moreover, PSI was at least 60 in 9/16 animals, and none had a SR higher than 0. Thus, these parameters were excluded from further analyses. Significantly higher DAP was present in Group MP_2 compared to Group SL_2 at TP5 (p = 0.046). Significantly higher values were also found in Group MP_2 compared to Group SL_2 for SEF R at TP5 (p = 0.038) and for EMG at TP5 (p = 0.022) and TP10 (p = 0.042). No other significant differences among groups were present. Details are reported in the S2 Appendix.

## Discussion

In the present study, methylphenidate was shown to be ineffective in influencing recovery in pigs. Our results were confirmed by all the assessed variables, and both when the drug was administered at the discontinuation of propofol infusion and at extubation. Its use to counteract the effects of propofol cannot be supported, at least at the doses used.

Due to the possible life-threatening and economic consequences of prolonged recovery periods, the interest in psychostimulants to enhance emergence from general anaesthesia has recently found a renewal of interest. Methylphenidate has been the most promising drug in this regard. Unfortunately, despite various studies have been started in the last years, not all the results have been published yet (e.g., ClinicalTrials.gov Identifiers: NCT02051452; NCT03610282).

The findings of the present research on recovery time and regain of protective reflexes are in contrast with most of the previous reports, in both humans and animals [5, 6, 16, 17]. Our clinical observations were mirrored by the monitored EEG activity. Indeed, SedLine® EEG-related variables showed no relevant differences among groups. Being the SedLine® validated for humans, we cannot exclude that differences were not found due to possible miscalculations [18]. However, no differences were present in EEG power, supporting the hypothesis that MP is not effective in stimulating cerebral activity and shortening recovery time at the end of propofol anaesthesia in this species, at least at the doses used.

Respiratory, cardiovascular and muscle activity modifications have been reported following MP administration. While in previous studies MP seemed to help maintaining a normal respiratory drive [11, 19], we did not observe any difference among groups, both when MP was administered at a deep level of anaesthesia and after extubation. Mild cardiovascular effects were noticed instead. Increase in DAP in experiment 1 and in DAP and MAP in experiment 2 were found, but not to a clinically relevant extent. This is in line with previous literature, often describing a mild rise in blood pressure following MP administration [16, 17, 20]. On the contrary, we did not observe any difference in heart rate. In this regard, contrasting results have been reported previously [11, 13, 19], and the development of life-threating arrhythmias following MP administration has been sporadically observed [21]. However, in the latter study, anaesthesia was maintained with halothane, drug known for its arrhythmogenic properties.

In the present trials, a mild increase in EMG activity was noticed in experiment 2, but it is unlikely to be of clinical relevance. This is in contrast to previous finding in humans, reporting MP to prevent muscular tremor and incoordination after anaesthesia [16, 21].

Several reasons might explain the differences between our results and previous findings.

First, different doses were used. In the present research, much higher doses were administered compared to previous reports. In the majority of the studies in animals, 1 to 5 mg/kg has been used, and much lower doses have been commonly administered in humans [6, 16, 19]. The doses of 10 and 20 mg/kg were chosen to increase the probability of detecting a clinically relevant response. Due to the lack of effect observed in experiment 1, the highest dose was

selected (20 mg/kg) for experiment 2. However, also in this case, no relevant effects were noticed.

Second, different administration protocols have been used. Methylphenidate has been commonly injected either during a continuous administration of a general anaesthetic or at its end [5, 6, 16, 17]. In experiment 1, no effects were noticed when MP was administered at the end of propofol infusion. Thus, it was administered immediately after extubation in experiment 2, targeting lower propofol plasmatic concentrations. Since no differences were found in both experiments, we do not expect dissimilar results from injecting MP during a continuous anaesthetic administration. Moreover, it would also meet less relevance for clinical practice.

Third, age could also have influenced the outcomes. Similar responses to MP have been described in young and geriatric rats [5]. In the present study, only juvenile pigs were included. Further researches should be conducted to clarify if age has an impact on the effect of MP.

Fourth, a species-specific responsiveness cannot be excluded. Different results have been found in various species [6, 7, 19] and, to the author knowledge, this is the first study investigating the effect of MP on recovery characteristics in pigs. Our results suggest that MP is not useful at the present dose to improve the recovery from propofol anaesthesia in this species.

Fifth, the anaesthetic used and the administration regimen could have influenced the MP effect. Anaesthesia induced by different injectable (e.g., barbiturate, propofol) or inhalant (e.g., isoflurane) anaesthetics has been counteracted by MP administration [5, 11, 17]. Even if direct comparison among anaesthetic protocols has never been performed, the information published so far does not seem to suggest an essential difference. Indeed, since MP stimulates an arousal pathway and do not specifically target a particular anaesthetic drug-receptor interaction, the influence of the anaesthetic drugs used is not expected to be highly relevant.

The present study has some limitations. First, raw EEG data were analysed only from the R-mid electrode. However, based on our previous analyses (unpublished data), EEG power was higher at this location compared to the more caudal one, and more pronounced differences were expected. Second, the MP doses chosen could have been too low to elicit any effect in pigs. However, they were much higher than reported in other species, making this possibility less likely. Third, the arterial partial pressure of carbon dioxide was not routinely assessed. Further studies should be conducted to better clarify the potential influence of carbon dioxide levels on sedation and EEG power in pigs.

## Conclusion

Methylphenidate does not shorten or modify anaesthesia recovery in pigs receiving propofol. Cardiovascular parameters seem to be only marginally affected.

## Supporting information

**S1 Appendix. The results of the variables assessed during experiment 1, for each group and time point, are reported, together with their graphical representation.** The blank spaces correspond to missing values. TP: time point; MP: methylphenidate; BSL: baseline; PSI: patient state index; NWR: nociceptive withdrawal reflex.
(PDF)

**S2 Appendix. The results of the variables assessed during experiment 2, for each group and time point, are reported, together with their graphical representation.** The blank spaces correspond to missing values. TP: time point; BSL: baseline; PSI: patient state index; NWR: nociceptive withdrawal reflex.
(PDF)

## Author Contributions

**Conceptualization:** Alessandro Mirra, Claudia Spadavecchia, Olivier Louis Levionnois.

**Data curation:** Alessandro Mirra, Fabiana Micieli, Mirjam Arnold.

**Formal analysis:** Alessandro Mirra.

**Funding acquisition:** Alessandro Mirra, Claudia Spadavecchia, Olivier Louis Levionnois.

**Investigation:** Alessandro Mirra, Fabiana Micieli, Mirjam Arnold, Claudia Spadavecchia, Olivier Louis Levionnois.

**Methodology:** Alessandro Mirra, Claudia Spadavecchia, Olivier Louis Levionnois.

**Project administration:** Claudia Spadavecchia, Olivier Louis Levionnois.

**Resources:** Claudia Spadavecchia, Olivier Louis Levionnois.

**Supervision:** Claudia Spadavecchia, Olivier Louis Levionnois.

**Validation:** Fabiana Micieli, Mirjam Arnold, Claudia Spadavecchia, Olivier Louis Levionnois.

**Visualization:** Claudia Spadavecchia, Olivier Louis Levionnois.

**Writing – original draft:** Alessandro Mirra.

**Writing – review & editing:** Fabiana Micieli, Mirjam Arnold, Claudia Spadavecchia, Olivier Louis Levionnois.

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
