## [Decision Letter · Decision Letter 0]

19 Dec 2023

PONE-D-23-36054The effect of Methylphenidate on anaesthesia recovery: an experimental study in pigs.PLOS ONE

Dear Dr. Mirra,

Thank you for submitting your manuscript to PLOS ONE. After careful consideration, we feel that it has merit but does not fully meet PLOS ONE’s publication criteria as it currently stands. Therefore, we invite you to submit a revised version of the manuscript that addresses the points raised during the review process.

**please carefully assess all the reviewers comments**

We look forward to receiving your revised manuscript.

Kind regards,

Silvia Fiorelli

Academic Editor

PLOS ONE

Journal Requirements:

This work was supported by the Swiss National Science Foundation (Spark) [https://www.snf.ch/en. Grant number CRSK-3_190926/1. Main grant applicant: OL], the Berne University Research Foundation and BEKB Förderfonds [https://forschungsstiftung.ch/. Grant number 22/2020. Main grant applicant: OL] and by the Faculty Clinical Research Platform (FKFP) of the Vetsuisse Faculty of the University of Bern [Main grant applicant: AM]. The funder had no role in study design, data collection and analysis, decision to publish, or preparation of the manuscript. There was no additional external funding received for this study.

Reviewers' comments:

Reviewer's Responses to Questions

**Comments to the Author**

1. Is the manuscript technically sound, and do the data support the conclusions?

Reviewer #1: Partly

Reviewer #2: Yes

2. Has the statistical analysis been performed appropriately and rigorously? 

Reviewer #1: Yes

Reviewer #2: Yes

3. Have the authors made all data underlying the findings in their manuscript fully available?

Reviewer #1: No

Reviewer #2: Yes

4. Is the manuscript presented in an intelligible fashion and written in standard English?

Reviewer #1: Yes

Reviewer #2: Yes

5. Review Comments to the Author

Reviewer #1: This research involves experimental studies on pigs to investigate the effects of methylphenidate on anesthesia recovery. While the study is novel, there are some concerns.

1. What criteria are used to determine the timing of intubation？

2. Please provide a clear description of the randomization method. A pig is anesthetized three times, is the method of treatment randomly assigned each time？Is it divided into three groups, taking all three anesthesia of the pigs as a total and then grouping them according to treatment？

3. As each pig undergoes anesthesia three times, how is it determined whether there is a mutual influence between the three anesthesia sessions?

4. The anesthesia protocols for Experiment 1 and Experiment 2 are different. Why not adopt the same protocol? Does the optimal result from Experiment 1 apply to Experiment 2?

5. In the appendix, what do the rows and columns represent? Please provide clear annotations.

6. Based on what standard was the dosage of drugs for pigs determined?

7. What is the primary outcome, and based on what indicators is the impact on anesthesia recovery judged?

Reviewer #2: I congratulate the authors for the clarity of the protocol description and the very good standard English.

I would clarify a little bit more the management of the extubation of the pigs that did not recover yet completely and that received Methylphenidate after extubation.

Thank you

6. PLOS authors have the option to publish the peer review history of their article (what does this mean?). If published, this will include your full peer review and any attached files.

Reviewer #1: No

Reviewer #2: **Yes: **Nora Di Tomasso

---

## [Author Response · Author response to Decision Letter 0]

23 Jan 2024

RESPONSE TO REVIEWERS

Dear reviewers,

First of all, thank you for the time spent for this manuscript.

Please find below the answers to your comments.

Reviewer #1

1. What criteria are used to determine the timing of intubation？

In veterinary medicine, the right moment is chosen based on clinical signs during attempts to intubate: lack of palpebral reflex, lack of jaw tone and laryngeal reflex. These criteria were also applied for the present study. Neuromuscular blocking agents are rarely used for the intubation.

2. Please provide a clear description of the randomization method. A pig is anesthetized three times, is the method of treatment randomly assigned each time？Is it divided into three groups, taking all three anesthesia of the pigs as a total and then grouping them according to treatment？

Each pig had to undergo three different treatments; thus, we applied a cross-over design. During each experimental day, one single treatment was administered.

The randomization was done a-priori,,before the start of the experiment. 

Between treatments (thus, between each experimental day), a wash-out period of at least 36 hours was allowed. A sentence has been added to clarify it (“Treatments and statistical analysis” paragraph).

3. As each pig undergoes anesthesia three times, how is it determined whether there is a mutual influence between the three anesthesia sessions?

When cross-over designs are applied, it is difficult to have an objective measure to confirm that absolutely no influence was present between the anaesthesia sessions. For this reason, a wash-out period is always ensured. In our case, at least 36 hours were waited, as similar time spans have been widely applied in the literature for cross-over designs. Moreover, we were using drugs (Propofol and Methylphenidate) that have a short half-life.

4. The anesthesia protocols for Experiment 1 and Experiment 2 are different. Why not adopt the same protocol? Does the optimal result from Experiment 1 apply to Experiment 2?

Thank you for pointing it out. As reported in the Discussion section: “In experiment 1, no effects were noticed when MP was administered at the end of propofol infusion. Thus, it was administered immediately after extubation in experiment 2, targeting lower propofol plasmatic concentrations.». Since the results obtained in Exp 1 did not show any effect of MP on recovery characteristics, we aimed at assessing if a lower plasmatic concentration would induce a different result. However, this was not the case.

If you are referring to the different target of suppression ratio (10-30% versus >80%), this was done for reason unrelated to the present study. In this context, it would have not been meaningful to administer MP with a SR>80% (second experiment) since we had already obtained no effect at a lighter anaesthesia level (10% <SR<30%; first experiment). Moreover, since the administration of MP was done using a clinical target (extubation), we did not expect a strong influence from the anaesthetic level at which propofol infusion was stopped.

5. In the appendix, what do the rows and columns represent? Please provide clear annotations.

Thank you for noticing it. Annotations have been added.

6. Based on what standard was the dosage of drugs for pigs determined?

The propofol doses were based on literature/PK models and personal experience. The methylphenidate doses were higher than those reported in the literature in other species for two reasons: 1) to target a stronger clinical effect; 2) because pigs commonly require higher doses of anaesthetics compared to humans or other mammals.

7. What is the primary outcome, and based on what indicators is the impact on anesthesia recovery judged?

We used both clinical (e.g., extubation, palpebral reflex, jaw tone) and instrumental monitorings (e.g., EEG, NWR) to detect changes. For this reason, we did not state one single primary outcome. We modify the text to make it clearer.

Reviewer #2

I would clarify a little bit more the management of the extubation of the pigs that did not recover yet completely and that received Methylphenidate after extubation.

In the experiment II you administered the methylphenidate after extubation. How were you able to extubate the pigs without they did not recover completely from sedation? 

Thank you for your comment. The extubation was performed when it was considered appropriate based on the clinical judgement (as it would be done for a clinical case). In veterinary medicine, when the animals do not tolerate the endotracheal tube anymore, extubation is performed. This was the case in the present study (as reported in line 201-202; a short clarification has been added in the text). After extubation, the animals are often still showing sedation. While this is generally uneventful, it can also lead to complications (mostly respiratory obstruction). Thus, we wanted to assess if methylphenidate could help in shortening this part of the recovery phase.

Why did you choose not to use miorelaxants for tracheal intubation and eventually the antagonist of them?

In veterinary medicine, it is uncommon to use myorelaxant before intubation. Even if we agree that it could have been an option, it was not deemed necessary, it would have increased the cost, and it would not reflect the common practice.

I found the dose of propofol used very high (20mg/kg/h + 4-6mg/kg bolus+ smaller dose boluses+ increased infusion of 6mg/kg/h). Could the use of such a dose of propofol be influenced the efficacy of methylphenidate?

Yes, high doses of propofol may influence the efficacy of Methylphenidate. However, it is reported in the literature and supported by our experience that pigs require higher doses of propofol compared to humans or other mammals. This is reinforced here by absence of sedative and other drugs. Moreover, the dose administered was adjusted and increased clinically to target clinical endpoints (endotracheal intubation, reaction to stimulation, EEG suppression) excluding the risk of administering unnecessarily high doses. Finally, the second part of the experiment was purposely designed to administer MP at lower plasma concentrations of propofol (at extubation compared to end of infusion) to investigate if this would induce a difference.

Did you test the CO2 (carbon dioxide)? In order to exclude a high-CO2 related sedation?

Thank you for clarifying.

Thank you for your comment. We did not perform routine blood gases to assess arterial CO2 partial pressure. We do not expect it to strongly influence sedation at the end -tidal concentrations had (also while considering a large dead space); on the other side, it could have influenced the EEG power (even if only few information has been reported in pigs). Further studies on the topic should be conducted. Following your comment, a limitation has been added in the text.

---

## [Decision Letter · Decision Letter 1]

22 Feb 2024

PONE-D-23-36054R1The effect of Methylphenidate on anaesthesia recovery: an experimental study in pigs.PLOS ONE

Dear Dr. Mirra,

Thank you for submitting your manuscript to PLOS ONE. After careful consideration, we feel that it has merit but does not fully meet PLOS ONE’s publication criteria as it currently stands. Therefore, we invite you to submit a revised version of the manuscript that addresses the points raised during the review process.

We look forward to receiving your revised manuscript.

Kind regards,

Silvia Fiorelli

Academic Editor

PLOS ONE

Journal Requirements:

**Additional Editor Comments:**

**ACADEMIC EDITOR:**

please carefully assess all the reviewers comments

 and consider to add this topic in the limit section

Reviewers' comments:

Reviewer's Responses to Questions

**Comments to the Author**

1. If the authors have adequately addressed your comments raised in a previous round of review and you feel that this manuscript is now acceptable for publication, you may indicate that here to bypass the “Comments to the Author” section, enter your conflict of interest statement in the “Confidential to Editor” section, and submit your "Accept" recommendation.

Reviewer #1: All comments have been addressed

Reviewer #2: All comments have been addressed

2. Is the manuscript technically sound, and do the data support the conclusions?

Reviewer #1: Yes

Reviewer #2: Yes

3. Has the statistical analysis been performed appropriately and rigorously? 

Reviewer #1: Yes

Reviewer #2: Yes

4. Have the authors made all data underlying the findings in their manuscript fully available?

Reviewer #1: Yes

Reviewer #2: Yes

5. Is the manuscript presented in an intelligible fashion and written in standard English?

Reviewer #1: Yes

Reviewer #2: Yes

6. Review Comments to the Author

Reviewer #1: The author has made revisions in response to the review comments, which is a positive development. However, despite the experimental results indicating that methylphenidate does not significantly affect the recovery process of pigs under propofol anesthesia, the clinical implications remain unclear.

Reviewer #2: (No Response)

7. PLOS authors have the option to publish the peer review history of their article (what does this mean?). If published, this will include your full peer review and any attached files.

Reviewer #1: No

Reviewer #2: No

---

## [Author Response · Author response to Decision Letter 1]

26 Mar 2024

Good morning,

I am sending the revised manuscript. Following the reviewer/editor comment, a sentence has been added at the beginning of the Discussion section.

Thank you very much for your time.

Best regards.

---

## [Editor Report · Decision Letter 2]

28 Mar 2024

The effect of Methylphenidate on anaesthesia recovery: an experimental study in pigs.

PONE-D-23-36054R2

Dear Dr. Alessandro Mirra,

We’re pleased to inform you that your manuscript has been judged scientifically suitable for publication and will be formally accepted for publication once it meets all outstanding technical requirements.

Kind regards,

Silvia Fiorelli

Academic Editor

PLOS ONE

Additional Editor Comments (optional):

Congratulations to the authors and thanks to the reviewers for the provided suggestions which really helped improve the quality of the manuscript
---

## [Editor Report · Acceptance letter]

3 Apr 2024

PONE-D-23-36054R2 

PLOS ONE

Dear Dr. Mirra, 

I'm pleased to inform you that your manuscript has been deemed suitable for publication in PLOS ONE. Congratulations! Your manuscript is now being handed over to our production team.

Kind regards, 

on behalf of

Dr. Silvia Fiorelli 

Academic Editor

PLOS ONE